# The *Caenorhabditis elegans* Tubby homolog dynamically modulates olfactory cilia membrane morphogenesis and phospholipid composition

Danielle DiTirro[†], Alison Philbrook[†], Kendrick Rubino, Piali Sengupta*

Department of Biology, Brandeis University, Waltham, United States

**Abstract** Plasticity in sensory signaling is partly mediated via regulated trafficking of signaling molecules to and from primary cilia. Tubby-related proteins regulate ciliary protein transport; however, their roles in remodeling cilia properties are not fully understood. We find that the *C. elegans* TUB-1 Tubby homolog regulates membrane morphogenesis and signaling protein transport in specialized sensory cilia. In particular, TUB-1 is essential for sensory signaling-dependent reshaping of olfactory cilia morphology. We show that compromised sensory signaling alters cilia membrane phosphoinositide composition via TUB-1-dependent trafficking of a PIP5 kinase. TUB-1 regulates localization of this lipid kinase at the cilia base in part via localization of the AP-2 adaptor complex subunit DPY-23. Our results describe new functions for Tubby proteins in the dynamic regulation of cilia membrane lipid composition, morphology, and signaling protein content, and suggest that this conserved family of proteins plays a critical role in mediating cilia structural and functional plasticity.

DOI: https://doi.org/10.7554/eLife.48789.001

*For correspondence:
sengupta@brandeis.edu

[†]These authors contributed equally to this work

## Introduction

Primary cilia are sensory organelles that are specialized to detect and transduce external stimuli (*Bangs and Anderson, 2017*; *Hilgendorf et al., 2016*; *May-Simera et al., 2017*). Cilia contain a microtubule-based axoneme surrounded by a membrane that houses cell- and context-specific subsets of signal transduction molecules. The axoneme is built by the highly conserved process of intraflagellar transport (IFT) that traffics molecules such as tubulin and signaling proteins into and out of cilia via kinesin and dynein molecular motors (*Goetz et al., 2009*; *Kozminski et al., 1993*; *Rosenbaum and Witman, 2002*; *Scholey, 2003*; *Taub and Liu, 2016*). IFT mechanisms have been extensively studied across species, and both core and accessory IFT molecules have been identified and characterized (*Cole et al., 1998*; *Scholey, 2003*; *Taschner and Lorentzen, 2016*). However, the pathways necessary for cilia membrane biogenesis, and trafficking of ciliary signaling molecules, remain to be fully described.

Cilia membrane volume and signaling protein content is regulated dynamically both as a function of cell type and external stimuli (*Doroquez et al., 2014*; *Falk et al., 2015*; *Goetz et al., 2009*; *Mesland et al., 1980*; *Mukhopadhyay et al., 2008*; *Mykytyn and Askwith, 2017*; *Silverman and Leroux, 2009*). For instance, vertebrate photoreceptor outer segments are built and maintained by cell-specific membrane delivery and retrieval mechanisms (*Goldberg et al., 2016*; *Molday and Moritz, 2015*). In mammalian cells, Sonic Hedgehog (Shh) signaling regulates ciliary trafficking of the Smoothened and GPR161 G-protein-coupled receptors (GPCRs), as well as the Shh receptor Patched (*Bangs and Anderson, 2017*; *Corbit et al., 2005*; *Huangfu et al., 2003*; *Mukhopadhyay et al., 2013*; *Rohatgi et al., 2007*). Since there is no protein translation within cilia,

bulk cilia membrane and associated proteins are delivered to the cilia base via vesicular trafficking prior to transport into cilia (*Jensen and Leroux, 2017*; *Mukhopadhyay et al., 2017*; *Nachury and Mick, 2019*). In one pathway, post-Golgi vesicles or recycling endosomes fuse to the membrane of a specialized domain at the cilium base referred to as the ciliary pocket or periciliary membrane compartment (PCMC) (*Benmerah, 2013*; *Ghossoub et al., 2011*; *Kaplan et al., 2012*). Vesicle fusion is followed by lateral diffusion, or more commonly IFT-mediated trafficking, of these proteins into the cilia membrane (*Badgandi et al., 2017*; *Fu et al., 2016*; *Hirano et al., 2017*; *Milenkovic et al., 2009*; *Mukhopadhyay et al., 2010*; *Picariello et al., 2019*). Removal of ciliary membrane proteins is mediated via ectocytosis from the cilia tip, endocytosis at the cilia base, or lateral diffusion (*Besharse et al., 1977*; *Clement et al., 2013*; *Lechtreck et al., 2009*; *Nager et al., 2017*; *Pal et al., 2016*; *Ye et al., 2018*). Cilia membrane delivery and retrieval mechanisms are tightly controlled to ensure effective and efficient signal transduction.

Members of the conserved family of Tubby (TUB) and Tubby-like (TULP) proteins link vesicular trafficking at the cilia base with IFT-mediated trafficking of membrane proteins within cilia (*Mukhopadhyay and Jackson, 2011*; *Wang et al., 2018*). TULP1 is expressed specifically in photoreceptors and regulates vesicular transport of phototransduction molecules in the inner segment, but does not travel into the photoreceptor cilium itself (*Hagstrom et al., 2001*; *Hagstrom et al., 1999*; *Milam et al., 2000*; *North et al., 1997*). In contrast, the more broadly expressed TULP3 and TUB proteins transport GPCRs and channels into cilia via interactions with the IFT-A complex (*Badgandi et al., 2017*; *Loktev and Jackson, 2013*; *Mukhopadhyay et al., 2010*; *Sun et al., 2012*). Similarly, the Tubby homologs dTULP in *Drosophila melanogaster* and TUB-1 in *Caenorhabditis elegans* mediate ciliary trafficking of channels and GPCRs in ciliated sensory neurons (*Brear et al., 2014*; *Mak and Ruvkun, 2004*; *Mukhopadhyay et al., 2005*; *Park et al., 2013*). Thus, TUB/TULP proteins represent excellent targets of regulatory pathways that dynamically modulate ciliary membrane protein trafficking.

All TUB/TULP proteins directly bind phosphatidylinositol 4,5-bisphosphate [$PI(4,5)P_2$] via their conserved C-terminal Tubby domains (*Santagata et al., 2001*). Binding of this phosphoinositide facilitates interaction of TUB/TULP proteins with ciliary transmembrane proteins in the PCMC/ciliary pocket and subsequent transport via the IFT-A complex into cilia (*Badgandi et al., 2017*; *Mukhopadhyay et al., 2010*). The PCMC membrane is enriched in $PI(4,5)P_2$, consistent with a requirement of this lipid for exo- and endocytosis (*De Craene et al., 2017*; *Martin, 2012*; *Martin, 2015*; *Posor et al., 2015*). However, $PI(4,5)P_2$ is depleted from the cilia membrane due to the presence of the phosphoinositide 5-phosphatase INPP5e within cilia (*Chávez et al., 2015*; *Garcia-Gonzalo et al., 2015*; *Jensen et al., 2015*; *Park et al., 2015*). Absence of $PI(4,5)P_2$ in the cilia membrane is proposed to weaken interaction of TUB/TULP proteins with their cargo and promote cargo release (*Badgandi et al., 2017*). Consequently, precise regulation of membrane phosphoinositide composition is critical for cilia membrane morphogenesis and membrane protein trafficking. Whether modulation of cilia and PCMC membrane phosphoinositide composition underlies plasticity in cilia protein content is unknown.

As in other animals, cell-specific sets of signal transduction molecules are trafficked and localized to the cilia of the 12 sensory neuron pairs present in the bilateral amphid sense organs in the head of the *C. elegans* hermaphrodite (*Nguyen et al., 2014*; *Roayaie et al., 1998*; *Tobin et al., 2002*; *Troemel et al., 1995*; *Wojtyniak et al., 2013*). A notable feature of these sensory cilia is their diverse morphologies. Eight of the 12 sensory neuron pairs contain one or two rod-like cilia ('channel' cilia), whereas the AWA, AWB, and AWC sensory neurons contain cilia ('wing' cilia) with unique axonemal ultrastructures and specialized membrane morphologies (*Doroquez et al., 2014*; *Perkins et al., 1986*; *Ward et al., 1975*). The membrane morphologies of the wing but not channel cilia are subject to further remodeling based on sensory inputs (*Mukhopadhyay et al., 2008*). The mechanisms that regulate cilia membrane biogenesis during development, and dynamic remodeling in response to sensory signaling in the adult, are not fully characterized.

Here, we show that the *C. elegans* TUB-1 Tubby homolog is required for the biogenesis of specialized membrane morphologies in wing cilia. TUB-1 is also necessary for sensory signaling-dependent expansion of the membrane in AWB wing cilia. Intriguingly, compromised sensory signaling results in increased levels of $PI(4,5)P_2$ as well as TUB-1 within AWB cilia, suggesting that ciliary membrane phosphoinositide composition can be dynamically altered in response to sensory inputs. This altered ciliary phospholipid composition is mediated via TUB-1-dependent ciliary localization of the

type I phosphatidylinositol-4-phosphate 5-kinase (PIP5K) PPK-1. We further show that TUB-1 regulates PPK-1 and PI(4,5)P$_2$ distribution in AWB dendrites in part via localization of the DPY-23 AP-2 µ2 subunit of the clathrin adaptor complex at the PCMC. Our results identify the conserved TUB-1 Tubby protein as a key regulator of membrane biogenesis in specialized cilia, demonstrate that cilia membrane phosphoinositide composition is subject to dynamic modulation as a function of sensory input, and describe a new function for TUB-1 in regulating cilia membrane lipid and protein content via regulated trafficking of a lipid kinase.

## Results

### TUB-1 regulates cilia membrane morphogenesis in specialized wing cilia

To characterize potential ciliary functions of TUB-1, we surveyed the morphologies of individual amphid sensory neuron cilia in *tub-1* mutant adult hermaphrodites. Wild-type AWA cilia are complex and highly branched, whereas AWC cilia exhibit large wing-like structures ('fans') comprised of a flattened membrane surrounding splayed out axonemal microtubules (*Doroquez et al., 2014*; *Evans et al., 2006*; *Perkins et al., 1986*) (*Figure 1A*). Each AWB neuron has two cilia, containing short proximal axonemes and distal membraneous regions with few disorganized microtubules (*Figure 1A*) (*Doroquez et al., 2014*; *Mukhopadhyay et al., 2008*). AWA cilia complexity was reduced in *tub-1* mutants; we also observed branches emanating from the AWA PCMC in these animals (*Figure 1A*). The extent of the membraneous ciliary fans was markedly decreased in AWC, and AWB cilia were severely truncated, in *tub-1* mutants (*Figure 1A–B*). The mutant AWB cilia morphology phenotype was fully rescued upon expression of wild-type *tub-1*, but not human TULP1 or TULP3, sequences specifically in AWB (*Figure 1B*, *Figure 1—figure supplement 1*). These observations indicate that TUB-1 acts in a cell-specific manner to shape specialized wing cilia morphology.

The phenotypes of AWB and AWC cilia suggested that the membraneous expansions in these cilia are reduced or lost in *tub-1* mutants. To test this notion, we labeled the AWB cilia membrane and the axoneme via co-expression of a myristoylated tagRFP reporter and the GFP-tagged OSM-6 IFT-B component, respectively. As reported previously, OSM-6::GFP was largely restricted to a short proximal segment corresponding to the region containing the organized axoneme (*Mukhopadhyay et al., 2007b*), whereas membrane-associated myr-tagRFP was present throughout the AWB cilia in wild-type animals (*Figure 1C*). The distal myr-tagRFP-labeled membraneous regions were lost in *tub-1* mutants while OSM-6::GFP localization was unaltered (*Figure 1C*), indicating that the cilia membrane, but not axonemes, are affected upon loss of *tub-1* in AWB cilia.

Loss of dTULP and TULP1 in *Drosophila* rhabdomeric and mammalian ciliary photoreceptors, respectively, leads to progressive retinal degeneration (*Chen et al., 2012*; *Gu et al., 1998*; *Hagstrom et al., 1998*). We asked whether AWB ciliary membrane is also progressively lost in *tub-1* mutants. Ciliogenesis is initiated following amphid sensory neuron birth late in embryogenesis in *C. elegans* (*Nechipurenko et al., 2017*; *Serwas et al., 2017*). Interestingly, the lengths of AWB cilia in L1 larvae were similar in wild-type and *tub-1(nr2004)* animals (*Figure 1B*). However, while wild-type AWB cilia elongated through postembryonic stages into adulthood, AWB cilia lengths in *tub-1* mutants were not further altered (*Figure 1B*). We infer that TUB-1 is required to extend the distal membraneous regions of AWB cilia during postembryonic development.

Unlike in wing cilia, axonemes span the lengths of channel cilia (*Doroquez et al., 2014*; *Perkins et al., 1986*; *Ward et al., 1975*) (*Figure 1D*). Consistent with the notion that TUB-1 does not regulate axoneme elongation, the lengths of the single rod-like channel cilia of the ASK sensory neurons were unaltered in *tub-1* mutants, whereas the lengths of the ASH and ASI neuronal cilia were only slightly, albeit significantly, shortened in these animals (*Figure 1D–E*) (*Brear et al., 2014*). Knockout of TULP3 has also been recently reported to result in a slight but significant shortening of primary cilia in mammalian cells (*Han et al., 2019*). Together, these observations indicate that TUB-1 is necessary for elaboration of the cilia membrane in a subset of amphid sensory neurons in *C. elegans*.

### TUB-1 regulates localization of ciliary transmembrane proteins in AWB

Loss of cilia membrane is expected to be associated with defects in the localization of ciliary transmembrane proteins. We previously showed that TUB-1 is required for localization of ciliary GPCRs in

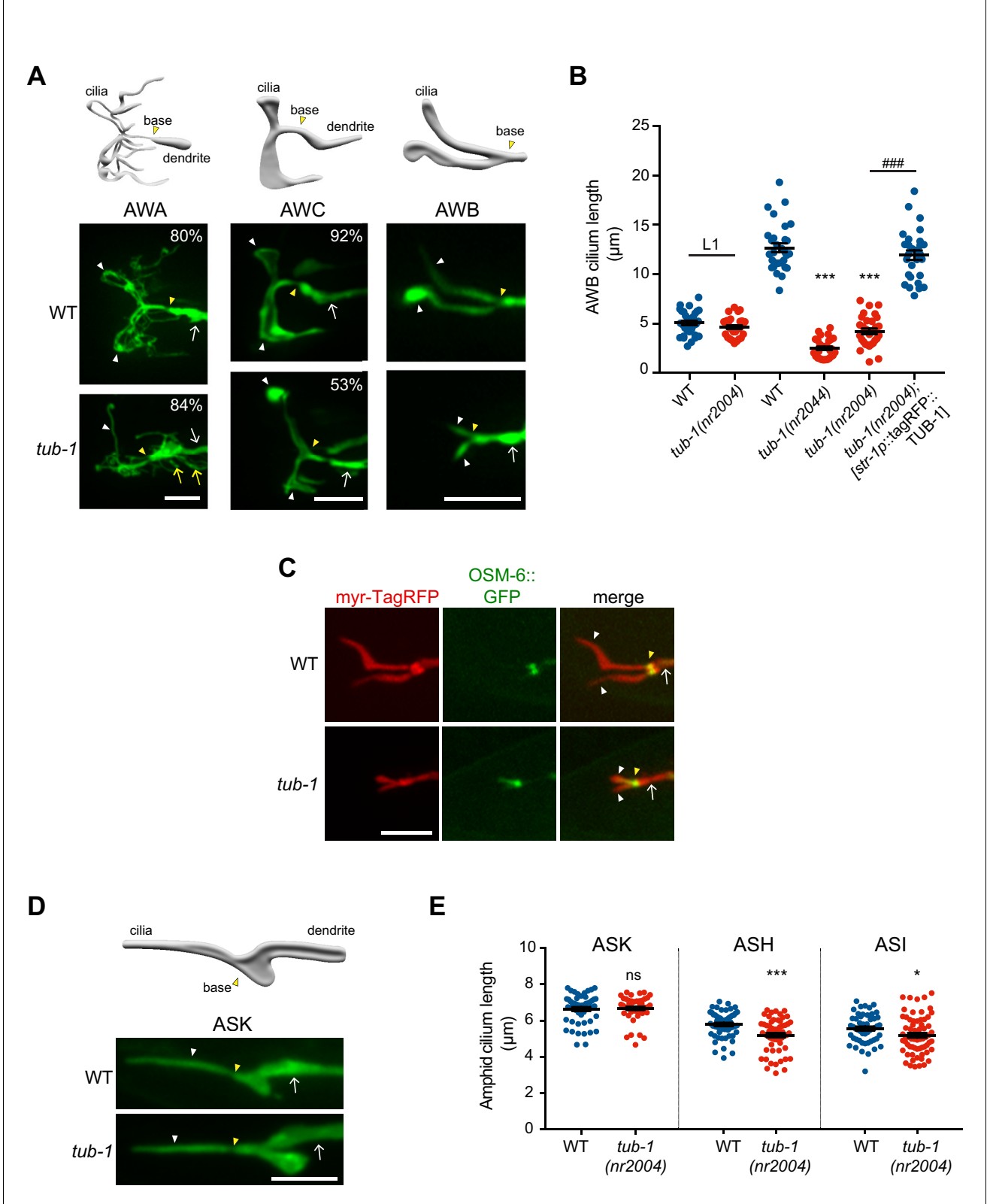

**Figure 1.** TUB-1 is necessary for membrane morphogenesis in wing cilia. (**A**) Representative images of AWA, AWC and AWB cilia in wild-type and *tub-1(nr2004)* mutants. Cartoons of cilia morphologies are shown at top. Numbers at top right (in AWA and AWC image panels) indicate the percentage of neurons exhibiting the shown phenotype; n > 45 neurons each. (**B**) Quantification of total AWB cilia lengths in animals of the indicated genotypes. Animals were adult hermaphrodites unless indicated otherwise. The *str-1* promoter drives expression primarily in AWB (*Troemel et al., 1997*). Each dot
*Figure 1 continued on next page*

*Figure 1 continued*

represents the combined AWB cilia lengths from a single neuron. *** and ### indicate different from wild-type or *tub-1* mutant respectively, at the comparable developmental stage at p<0.001 (ANOVA with Tukey's post-hoc test). (C) Representative images of wild-type and *tub-1(nr2004)* animals co-expressing the indicated fusion proteins. (D) (Left) Cartoon and representative images of ASK cilia in wild-type and *tub-1(nr2004)* mutants. (Right) Quantification of ASK, ASH, and ASI cilia lengths in adult hermaphrodites of the indicated genotypes. Each dot is cilia length from a single neuron. * and *** indicate different from wild-type at p<0.05 and 0.001, respectively (t-test); ns – not significant. In all images, yellow and white arrowheads indicate the cilia base and cilia, respectively; arrow indicates the dendrite. Scale bars: 5 μm. In all scatter plots, horizontal line is mean; error bars are SEM.

DOI: https://doi.org/10.7554/eLife.48789.002

The following source data and figure supplement are available for figure 1:

**Source data 1.** Data for *Figure 1B and E*, and *Figure 1—figure supplement 1*.

DOI: https://doi.org/10.7554/eLife.48789.004

**Figure supplement 1.** TUB-1 regulates AWB cilia morphology.

DOI: https://doi.org/10.7554/eLife.48789.003

AWB (*Brear et al., 2014*). In *tub-1* mutants, all previously examined GPCR::GFP fusion proteins are depleted from AWB cilia and are instead enriched at the PCMC (*Kaplan et al., 2012*). Although localization of a subset of GPCRs is also TUB-1-dependent in ASK, ciliary localization of the SRBC-64 GPCR is TUB-1-independent in this neuron type (*Brear et al., 2014*). To ask whether TUB-1 acts in a neuron- or GPCR-specific manner to localize SRBC-64, we examined whether SRBC-64 localization in AWB requires TUB-1. We found that SRBC-64::GFP was mislocalized to the PCMC in *tub-1* mutants (*Figure 2A*), similar to the localization defects of other examined GPCRs. These observations imply that TUB-1 is required to correctly traffic and/or localize multiple GPCRs to AWB cilia.

In addition to GPCRs, Tubby family proteins traffic multiple classes of ciliary transmembrane proteins in ciliated mammalian cells and *Drosophila* sensory neurons (*Badgandi et al., 2017*; *Grossman et al., 2011*; *Park et al., 2013*; *Park et al., 2015*). In *C. elegans,* the ciliary localization of the cyclic nucleotide-gated channel protein TAX-4 was also TUB-1-dependent in AWB, such that this protein was depleted from cilia and mislocalized to the PCMC in *tub-1* mutants (*Figure 2B*). The protein mislocalization phenotype in AWB is unlikely to simply be a secondary consequence of shortened cilia since TAX-4 localization was unaltered in *odr-3(gof)* Gα mutants in which AWB cilia are also severely shortened (*Figure 2—figure supplement 1*) (*Mukhopadhyay et al., 2008*). Recently, TULP3 was shown to regulate localization of the small GTPase ARL13b in mammalian cilia (*Han et al., 2019*; *Hwang et al., 2019*; *Legué and Liem, 2019*). In *C. elegans*, ARL-13 is also restricted to cilia, and regulates ciliary localization of multiple classes of transmembrane proteins (*Cevik et al., 2010*; *Li et al., 2010*; *Li et al., 2012*; *Nguyen et al., 2014*; *Wojtyniak et al., 2013*). While ARL-13::tagRFP was restricted to AWB cilia in wild-type animals, this fusion protein was also present at the AWB PCMC and distal dendritic regions in *tub-1* mutants (*Figure 2C*). Localization of the DYF-19 basal body and MKS-5 transition zone proteins (*Wei et al., 2013*; *Williams et al., 2011*) was unaltered in the AWB cilia of *tub-1* mutants (*Figure 2D–E*), suggesting that mislocalization of transmembrane and membrane-associated proteins in *tub-1* mutants was likely not due to gross defects in transition zone organization.

Since TUB-1 is also required to localize a subset of GPCRs in ASK channel cilia (*Brear et al., 2014*) despite playing only a minor role in regulating channel cilia length, we examined the localization of additional ciliary proteins in channel cilia in *tub-1* mutants. The TAX-2::GFP channel subunit was mislocalized to the distal dendritic ends of ASK in *tub-1* mutants, similar to the mislocalization phenotype of TAX-4::GFP in AWB (*Figure 2F*). However, in contrast to AWB, ARL-13::RFP localization was unaltered in *tub-1* mutants in ASH cilia (*Figure 2G*). These data suggest that the role of TUB-1 in regulating ciliary trafficking and localization of transmembrane and membrane-associated proteins in AWB and channel cilia are likely distinct.

## TUB-1 acts both at the PCMC and within cilia to shape AWB cilia morphology

To begin exploring the mechanisms by which TUB-1 regulates wing cilia membrane morphogenesis, we examined its subcellular localization in AWB. TUB-1 has been reported to be cytoplasmic and present throughout the cell including within the cilia of sensory neurons in *C. elegans* (*Mak et al.,*

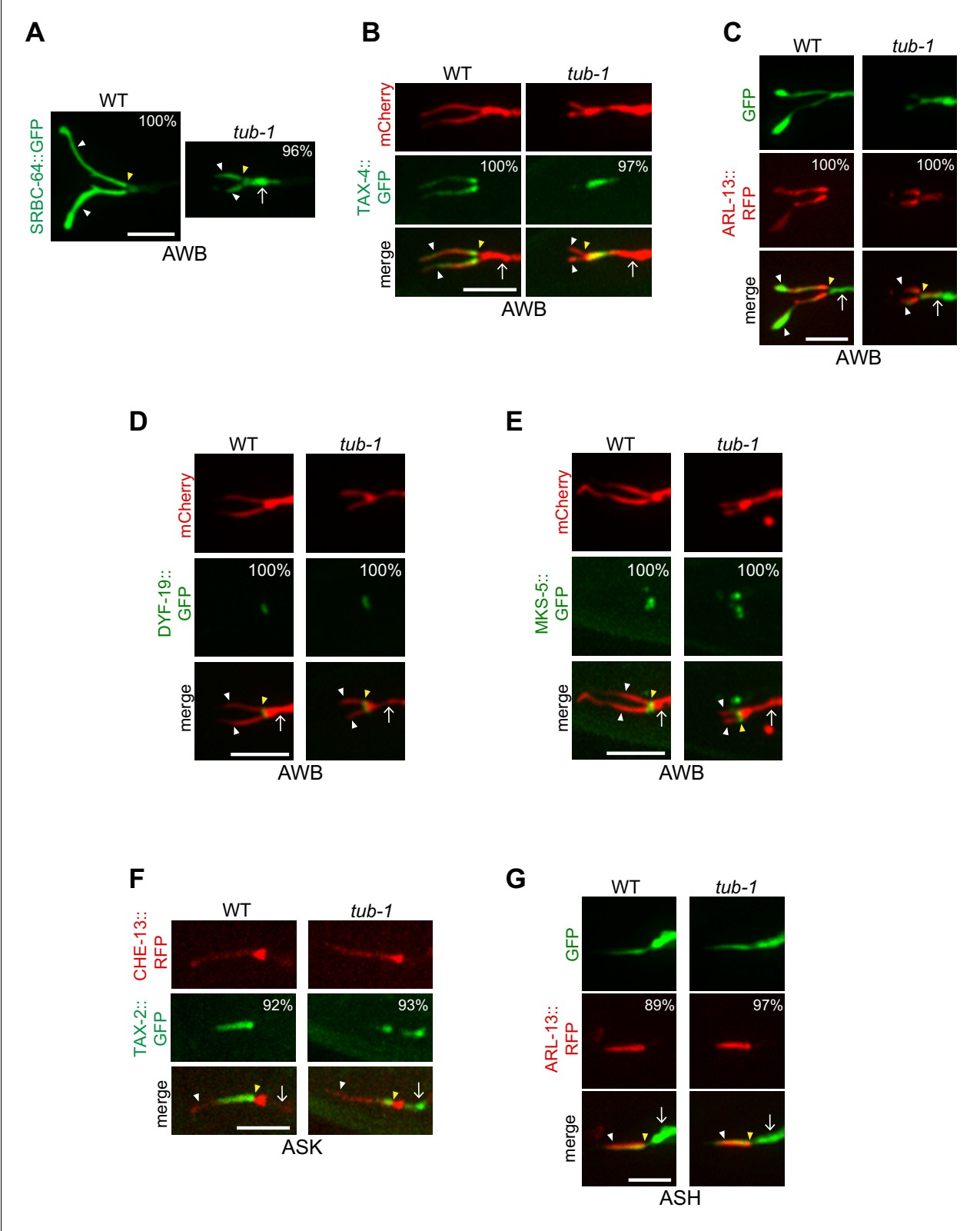

**Figure 2.** TUB-1 regulates ciliary targeting of transmembrane proteins in AWB. (A–G) Representative images of localization patterns of the indicated fusion proteins in wild-type (WT) and *tub-1(nr2004)* mutants in AWB (A–E), ASK (F) and ASH (G). The AWB, ASK, and ASH neurons and/or cilia are marked via expression of *str-1*p::mCherry (B, D, E), *str-1*p::GFP (C), *srbc-66*p::CHE-13::RFP (F), or *sra-6*p::GFP (G). Numbers at top right indicate percentage of animals exhibiting the phenotype; n ≥ 20 neurons each. Localization patterns were analyzed only in animals expressing the fusion

*Figure 2 continued on next page*

*Figure 2 continued*

protein. Approximately similar percentages of wild-type and mutant animals expressed the fusion protein with the exception of animals expressing TAX-4::GFP (83% and 68% of wild-type and *tub-1* mutants expressed TAX-4::GFP, respectively). Yellow and white arrowheads indicate the cilia base and cilia, respectively; arrow indicates the dendrite. Scale bars: 5 μm.

DOI: https://doi.org/10.7554/eLife.48789.005

The following figure supplement is available for figure 2:

**Figure supplement 1.** TAX-4 localization is unaltered in *odr-3(gof)* mutants.

DOI: https://doi.org/10.7554/eLife.48789.006

*2006*; *Mukhopadhyay et al., 2005*). Consistent with these observations, a functional tagRFP::TUB-1 fusion protein (*Figure 1B*) was present in the cytoplasm and excluded from the nucleus in AWB (*Figure 3A*). We noted marked enrichment of the protein at the PCMC with lower levels within the cilia proper (*Figure 3A*).

Tubby-like proteins contain a divergent N-terminal domain (NTD) and a conserved C-terminal Tubby domain (*Ikeda et al., 2002*; *Mukhopadhyay and Jackson, 2011*; *Wang et al., 2018*) (*Figure 3B*). Interaction of TULP3/TUB with diverse ciliary localization sequences present on ciliary transmembrane proteins is facilitated by binding of PI(4,5)P$_2$ via a pair of conserved positively charged residues in the Tubby domain (*Figure 3B*) (*Badgandi et al., 2017*; *Mukhopadhyay et al., 2010*; *Quinn et al., 2008*; *Santagata et al., 2001*). Residues in the NTDs of TULP3 and TUB interact with IFT-A complex proteins (*Figure 3B*) that in turn traffic the TULP3/TUB-associated transmembrane protein complex into cilia (*Badgandi et al., 2017*; *Mukhopadhyay et al., 2010*). Thus, both the NTD and Tubby domains are required for ciliary trafficking functions of TULP3/TUB in mammalian cells (*Badgandi et al., 2017*; *Mukhopadhyay et al., 2010*; *Park et al., 2013*).

Consistent with trafficking into the cilium being mediated by residues in the TUB-1 NTD, we found that the TUB-1 Tubby domain alone [tagRFP::TUB-1(CTD)] was excluded from AWB cilia and restricted to an expanded domain at the cilia base (*Figure 3C*, *Figure 3E*). In contrast, a fusion protein containing the NTD alone [tagRFP::TUB-1(NTD)] and lacking the PI(4,5)P$_2$-binding Tubby domain was no longer enriched at the PCMC and was present uniformly in the AWB dendrites and cilia (*Figure 3C*, *Figure 3E*). Expression of neither fusion protein alone was sufficient to rescue the AWB cilia phenotype of *tub-1* mutants (*Figure 3F*). These observations imply that TUB-1 functions at both the PCMC and within cilia to shape AWB cilia morphology.

We next characterized the residues within the NTD and Tubby domains that mediate TUB-1 localization to distinct regions in AWB. Residues in the NTDs of TULP3 and TUB that mediate interaction with IFT-A complex proteins were predicted to be absent in TUB-1 (*Mukhopadhyay and Jackson, 2011*). However, upon closer examination of the TUB-1 NTD sequence, we identified a subset of conserved residues in the homologous IFT-A binding domain (*Figure 3B*). Mutations in these residues resulted in depletion of the mutant protein [tagRFP::TUB-1(QRKRmut)] from cilia and localization to an expanded domain at the AWB ciliary base and distal dendritic region (*Figure 3C*, *Figure 3E*). This fusion protein also failed to rescue the AWB ciliary morphology defects of *tub-1* mutants (*Figure 3F*). Consistent with the notion that TUB-1 is trafficked via IFT into the AWB cilia, both *tub-1* and *daf-10* IFT122 IFT-A mutants exhibit similar phenotypes of ciliary GPCR accumulation at the AWB PCMC (*Brear et al., 2014*), and resulted in similar AWB cilia truncation phenotypes (*Figure 3—figure supplement 1A*). Localization of DAF-10 was unaltered in the AWB cilia of *tub-1* mutants (*Figure 3—figure supplement 1B*). However, while mutations in the *kap-1* kinesin II subunit required for IFT did not affect ciliary TUB-1 localization, we observed increased accumulation of TUB-1 at the PCMC (*Figure 3D*), further suggesting that TUB-1 is trafficked into cilia via IFT.

Mutating the predicted PI(4,5)P$_2$-binding residues (*Figure 3B*) in the Tubby domain of TUB-1 [tagRFP::TUB-1(KRmut)] resulted in loss of PCMC enrichment similar to the localization pattern of tagRFP::TUB-1(NTD) (*Figure 3C*, *Figure 3E*), and abolished the ability of this protein to rescue the AWB cilia phenotype of *tub-1* mutants (*Figure 3F*). Although Tubby proteins have also been reported to localize to the nucleus under specific conditions (*Boggon et al., 1999*; *He et al., 2000*; *Kim et al., 2017*; *Park et al., 2013*; *Santagata et al., 2001*), we did not detect nuclear localization of full-length TUB-1, TUB-1(NTD) or TUB-1(KRmut) fusion proteins (*Figure 3A*, *Figure 3—figure supplement 1C*). We infer that TUB-1 is trafficked into cilia via interaction of its NTD with the IFT

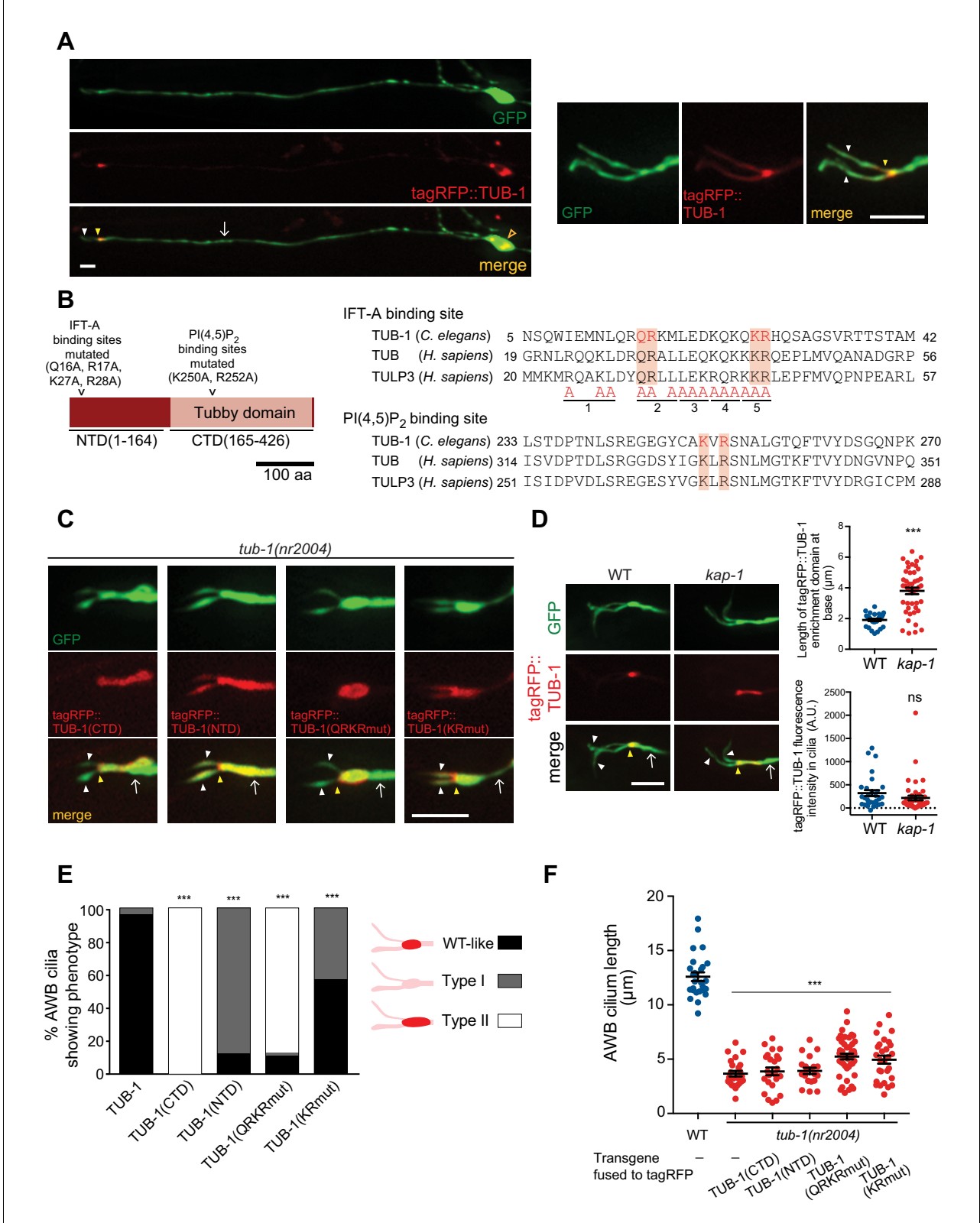

**Figure 3.** TUB-1 acts both at the AWB PCMC and within cilia to regulate AWB cilia morphology. (**A**) Representative images of tagRFP::TUB-1 localization in AWB indicating cytoplasmic localization in the soma (open orange arrow) and enrichment at the dendritic ends (yellow arrowheads; left and right images). tagRFP::TUB-1 is also present at lower levels in AWB cilia (white arrowheads; left and right images). The AWB neurons were visualized via expression of *str-1*p::GFP. Scale bars: 5 μm. (**B**) (Left) Cartoon of TUB-1 domains. Residues mutated in the IFT-A binding site in the

*Figure 3 continued on next page*

*Figure 3 continued*

N-terminal domain (NTD) and PI(4,5)P$_2$ binding site in the C-terminal Tubby domain (CTD) are indicated. (Right) Alignment of the predicted IFT-A binding domains in *C. elegans* TUB-1 and human TUB and TULP3 proteins. Residues in TULP3 which when replaced with Alanine abolishes IFT-A binding are shown; shown mutations in any of the five underlined regions abolish binding (*Mukhopadhyay et al., 2010*). Red shading indicates conserved residues mutated in this work. (C) Representative images of the indicated fusion proteins in *tub-1(nr2004)* mutants in AWB. AWB was visualized via expression of *str-1*p::GFP. (D) Representative images (left) and quantification of tagRFP::TUB-1 localization at the cilia base (top right) and cilia (bottom right) in wild-type and *kap-1(ok676)* mutants. The AWB neurons were visualized via expression of *str-1*p::GFP. *** indicates different from wild-type at p<0.001 (t-test); ns – not significant. (E) Percentage of AWB cilia exhibiting the indicated localization patterns of the shown fusion proteins in *tub-1(nr2004)* animals. Proteins were visualized via fusion with tagRFP. n ≥ 20 neurons each. *** indicates different from wild-type at p<0.001 (Fisher's exact test). (F) Quantification of total AWB cilia lengths in wild-type or *tub-1(nr2004)* animals expressing the indicated fusion proteins. *** indicates different from wild-type at p<0.001 (ANOVA with Tukey's post-hoc test). In all images, yellow and white arrowheads indicate the cilia base and cilia, respectively; arrow indicates the dendrite. Scale bars: 5 µm. In all scatter plots, each dot represents measurements from a single AWB neuron. Horizontal line is mean; error bars are SEM.

DOI: https://doi.org/10.7554/eLife.48789.007

The following source data and figure supplement are available for figure 3:

**Source data 1.** Data for *Figure 3D,E,F*, and *Figure 3—figure supplement 1A*.

DOI: https://doi.org/10.7554/eLife.48789.009

**Figure supplement 1.** *tub-1* shares a subset of phenotypes with *daf-10*.

DOI: https://doi.org/10.7554/eLife.48789.008

machinery, whereas enrichment of TUB-1 at the AWB PCMC requires interaction with PI(4,5)P$_2$ via its Tubby domain.

## TUB-1 is required for sensory signaling-dependent expansion of the AWB ciliary membrane

We previously showed that under conditions of reduced or absent sensory signaling, the ciliary membrane in AWB is significantly expanded resulting in a membraneous fan-like shape (*Mukhopadhyay et al., 2008*). Given a role for TUB-1 in regulating AWB cilia membrane morphogenesis during development, we tested whether TUB-1 is necessary for sensory signaling-dependent remodeling of AWB cilia.

As reported previously (*Mukhopadhyay et al., 2008*), AWB cilia width is significantly increased in *odr-1* receptor guanylyl cyclase signaling mutants (*Figure 4A*). This sensory signaling-dependent ciliary membrane expansion was fully suppressed upon loss of *tub-1* (*Figure 4A*). Loss-of-function mutations in *daf-10* also suppressed the expanded fan phenotype of *odr-1* mutants (*Figure 4—figure supplement 1*). The subcellular localization of Tubby proteins can be altered by cellular signaling (*Chen et al., 2012*; *Kim et al., 2014*; *Santagata et al., 2001*); we asked whether the expanded fan-like structure in *odr-1* mutants is accompanied by increased ciliary TUB-1 levels. Indeed, we found that *odr-1* mutants showed enrichment of a TUB-1 fusion protein in AWB cilia as compared to levels at the PCMC (*Figure 4B*). These results raise the possibility that reduced sensory signaling in *odr-1* mutants requires increased ciliary localization of TUB-1 to modulate ciliary membrane expansion. However, it remains possible that the severely truncated cilia in *tub-1* and *daf-10* mutants are unable to elaborate fans.

While TUB-1 has previously been suggested to undergo IFT in *C. elegans* sensory cilia (*Mukhopadhyay et al., 2005*), we were unable to perform live imaging of TUB-1 movement in wild-type AWB cilia due to low ciliary expression levels of this protein (*Figure 3A*) and the occurrence of IFT primarily in the short proximal region containing the axoneme (*Mukhopadhyay et al., 2007b*). To examine TUB-1 ciliary dynamics, we instead quantified recovery of GFP::TUB-1 fluorescence in the AWB cilia of wild-type and *odr-1* mutants following photobleaching (FRAP). We found that while fluorescence recovery occurred at a similar rate in wild-type and *odr-1* mutants, a significantly higher fraction of the TUB-1 fusion protein was mobile in the AWB cilia of *odr-1* mutants as compared to wild-type animals (*Figure 4C–E*). These results further support the notion that decreased sensory signaling in *odr-1* mutants increases trafficking of TUB-1 within AWB cilia.

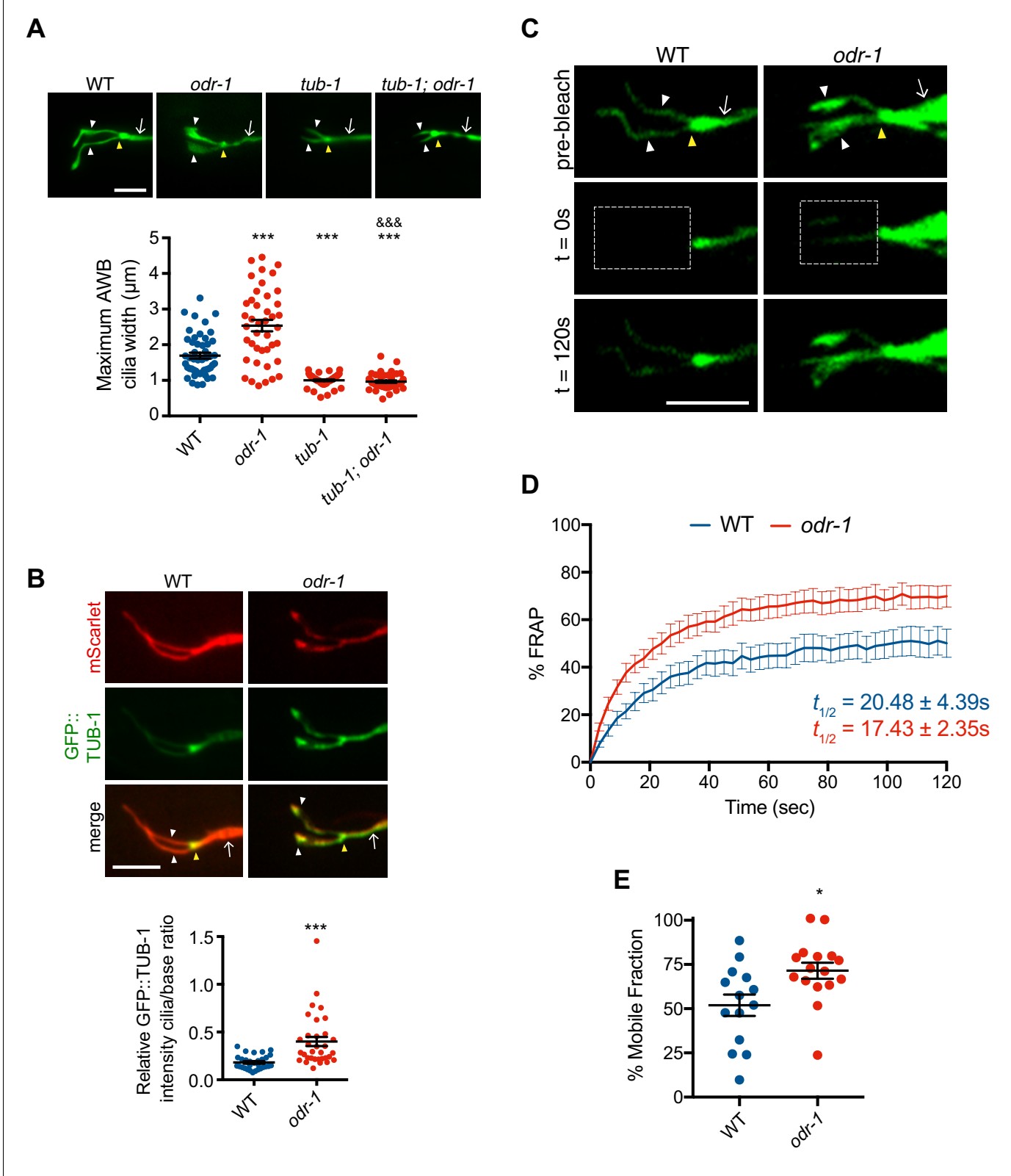

**Figure 4.** Reduced sensory signaling increases TUB-1 localization within AWB cilia. (**A**) (Top) Representative images of AWB cilia morphologies in animals of the indicated genotypes. Alleles used were *tub-1(nr2004)* and *odr-1(n1936)*. The AWB neurons were visualized via expression of *str-1*p::myr-GFP. (Bottom) Quantification of maximum AWB cilia widths. *** and &&& indicate different from wild-type and *odr-1*, respectively, at p<0.001 (ANOVA and post-hoc Tukey's test). (**B**) (Top) Representative images of GFP::TUB-1 localization in wild-type or *odr-1(n1936)* mutants. The AWB neurons were

*Figure 4 continued on next page*

*Figure 4 continued*

visualized via expression of *str-1*p::GFP::TUB-1::SL2::mScarlet. (Bottom) Quantification of relative GFP::TUB-1 fluorescence intensity within cilia vs the cilia base. *** indicates different from wild-type at p<0.001 (t-test). (**C**) Representative images of AWB expressing GFP::TUB-1 pre-bleach, and at 0 s and 120 s post-bleach. The bleached area is indicated by a dotted box. (**D**) Mean fluorescence recovery following photobleaching normalized to pre-photobleached intensity at 0 s in wild-type and *odr-1(n1936)* animals. Errors are SEM. n ≥ 14 animals; three independent experiments. (**E**) Percent mobile fraction of GFP::TUB-1 in AWB cilia in wild-type and *odr-1* animals calculated from data shown in D. * indicates different from wild-type at p<0.05 (t-test). In all images, yellow and white arrowheads indicate the cilia base and cilia, respectively; arrow indicates the dendrite. Scale bars: 5 μm. In all scatter plots, each dot represents measurements from a single AWB neuron. Horizontal line is mean; error bars are SEM.

DOI: https://doi.org/10.7554/eLife.48789.010

The following source data and figure supplement are available for figure 4:

**Source data 1.** Data for *Figure 4A,B,D,E*, and *Figure 4—figure supplement 1*.

DOI: https://doi.org/10.7554/eLife.48789.012

**Figure supplement 1.** Loss-of-function mutations in *daf-10* suppress the expanded 'fan' phenotype of *odr-1* mutants.

DOI: https://doi.org/10.7554/eLife.48789.011

## Ciliary membrane PI(4,5)P$_2$ levels are increased in *odr-1* signaling mutants

The increased presence of TUB-1 in the AWB cilia of *odr-1* mutants is reminiscent of phenotypes observed upon manipulation of ciliary membrane PI(4,5)P$_2$ levels. PI(4,5)P$_2$ is excluded from the cilia membrane in both vertebrates and *Drosophila*, primarily due to ciliary localization of the phosphatases such as INPP5e which converts PI(4,5)P$_2$ to PI(4)P (*Bielas et al., 2009*; *Chávez et al., 2015*; *Garcia-Gonzalo et al., 2015*; *Jacoby et al., 2009*; *Park et al., 2015*; *Prosseda et al., 2017*). Since TULP3/TUB proteins interact directly with PI(4,5)P$_2$, absence of ciliary PI(4,5)P$_2$ is associated with low levels of these proteins in wild-type cilia. However, upon depletion of INPP5e, increased ciliary PI(4,5)P$_2$ markedly enhances ciliary levels of TULP/TUB proteins (*Chávez et al., 2015*; *Garcia-Gonzalo et al., 2015*; *Park et al., 2015*). We asked whether increased ciliary localization of TUB-1 in *odr-1* mutants is associated with changes in ciliary membrane PI(4,5)P$_2$ distribution.

We first examined PI(4,5)P$_2$ distribution in wild-type AWB cilia using a GFP::PLCδ1-PH fusion protein (*Jensen et al., 2015*). Similar to observations in other organisms, *C. elegans* sensory cilia membranes are depleted in PI(4,5)P$_2$, although individual sensory cilia have not been examined (*Jensen et al., 2015*). We confirmed that GFP::PLCδ1-PH was associated with the dendritic and PCMC, but not the ciliary, membranes of AWB (*Figure 5A–B*, also see *Figure 6D–F*). In AWB dendrites, we observed that GFP::PLCδ1-PH exhibited maximal fluorescence intensity ~2–3 μm proximally from the cilia base (*Figure 5A*, also see *Figure 6D–E*).

We tested whether increasing ciliary PI(4,5)P$_2$ levels is sufficient to increase ciliary localization of TUB-1. The CIL-1 and INPP-1 phosphatases are the closest homologs of INPP5e in *C. elegans* (*Bae et al., 2009*). CIL-1 does not appear to regulate PI(4,5)P$_2$ levels (*Bae et al., 2009*). However, in *inpp-1* mutants, PLCδ1-PH was now also present in AWB cilia (*Figure 5A–B*). Consistent with the preferential association of TUB-1 with PI(4,5)P$_2$, we also observed higher levels of tagRFP::TUB-1 in AWB cilia in *inpp-1* mutants; this altered localization phenotype was rescued upon expression of a wild-type *inpp-1* cDNA (*Figure 5C*). We next asked whether similar to *inpp-1* mutants, ciliary PI(4,5)P$_2$ levels are also increased in *odr-1* mutants to account for the increased ciliary localization of TUB-1. Indeed, we found that the AWB cilia membrane contained higher levels of PI(4,5)P$_2$ relative to dendritic concentrations in *odr-1* mutants as compared to the ratio in wild-type animals (*Figure 5A–B*, also see *Figure 6D*, *Figure 6F*). We conclude that compromised sensory signaling in *odr-1* mutants is associated with increased ciliary PI(4,5)P$_2$ distribution, which in turn is correlated with increased ciliary TUB-1 localization. However, since AWB cilia did not exhibit a fan-like phenotype in *inpp-1* mutants (*Figure 5A*), increased ciliary PI(4,5)P$_2$ and TUB-1 levels are not sufficient to remodel AWB cilia morphology (see Discussion).

## Ciliary localization of the PPK-1 PIP5K is increased in *odr-1* signaling mutants

How might reduced sensory signaling increase ciliary PI(4,5)P$_2$? Reduced sensory signaling could result in depletion of ciliary INPP-1 and/or increased ciliary localization of a PIP5K that generates PI(4,5)P$_2$ from PI(4)P. Unlike in mammalian cells or in *Drosophila* (*Chávez et al., 2015*; *Garcia-*

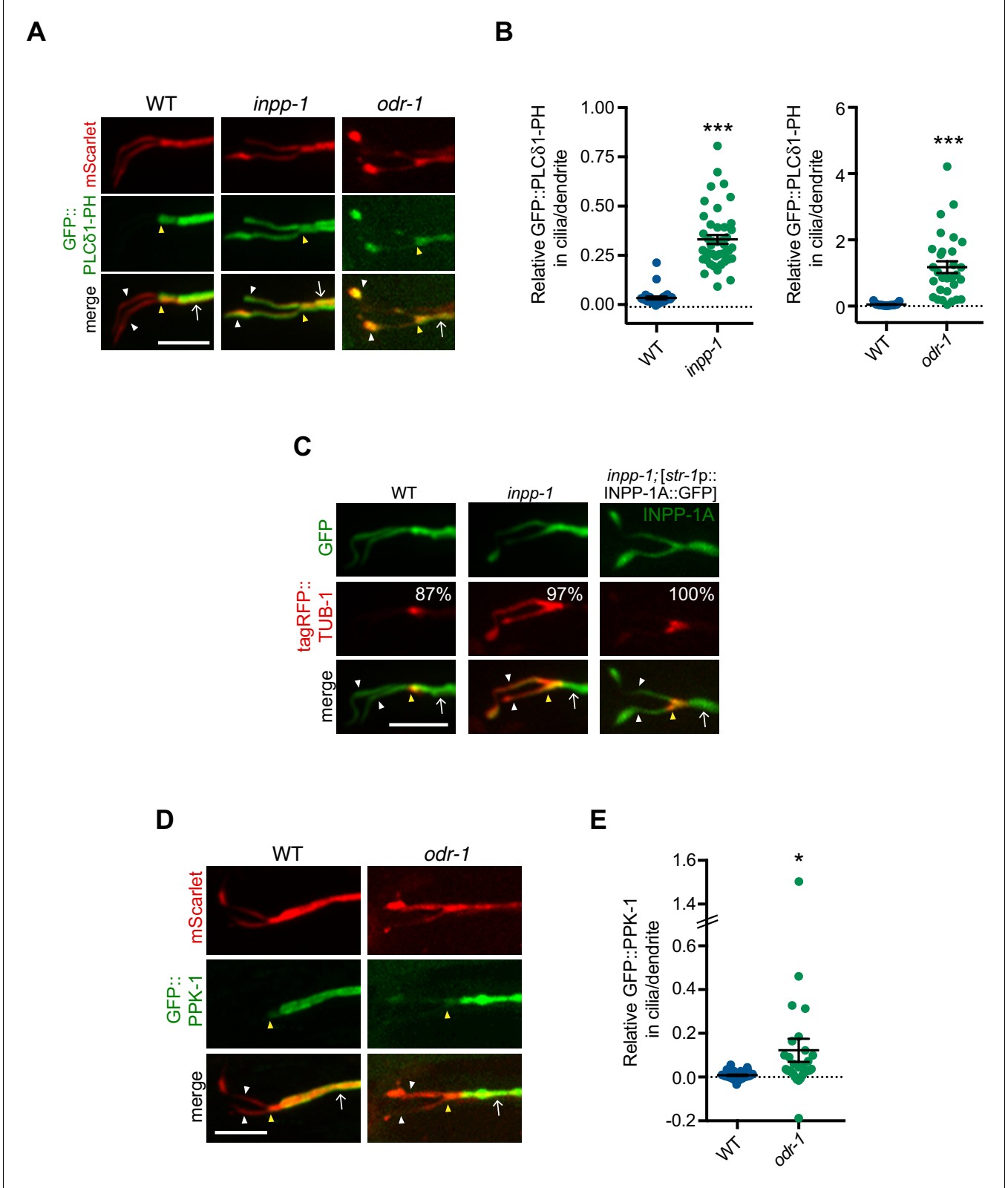

**Figure 5.** Levels of PI(4,5)P$_2$ and PPK-1 are increased within AWB cilia in *odr-1* mutants.  (A, C, D) Representative images of GFP::PLCδ1-PH (A), tagRFP::TUB-1 (C) and GFP::PPK-1 (D) localization in AWB cilia of animals of the indicated genotypes. Alleles used were *odr-1(n1936)* and *inpp-1 (gk3262)*. AWB neurons were visualized via expression of *str-1*p::GFP::PLCδ1-PH::SL2::mScarlet (A), *str-1*p::GFP (C), *str-1*p::INPP-1A::GFP (C, right column) or GFP::PPK-1::SL2::mScarlet (D). Numbers at top right in C indicate percentage of animals exhibiting the phenotype; n ≥ 23 neurons. Yellow
*Figure 5 continued on next page*

Figure 5 continued

and white arrowheads indicate the cilia base and cilia, respectively; arrow indicates the dendrite. Scale bars: 5 μm. (B, E) Quantification of relative GFP::PLCδ1-PH (B) and GFP::PPK-1 (E) fluorescence intensities in AWB cilia vs dendrites in animals of the indicated genotypes. * and *** indicate different from wild-type at $p<0.05$ and $p<0.001$, respectively (t-test). Each dot represents measurements from a single AWB neuron. Horizontal line is mean; error bars are SEM.

DOI: https://doi.org/10.7554/eLife.48789.013

The following source data and figure supplement are available for figure 5:

**Source data 1.** Data for *Figures 5B and E*, and *Figure 5—figure supplement 1*.
DOI: https://doi.org/10.7554/eLife.48789.015

**Figure supplement 1.** INPP-1 localization is unaltered in *tub-1* and *odr-1* mutants.
DOI: https://doi.org/10.7554/eLife.48789.014

*Gonzalo et al., 2015*; *Park et al., 2015*), a functional INPP-1::GFP fusion protein (*Figure 5C*) was present at similar levels throughout the dendrites and cilia of wild-type AWB neurons (*Figure 5—figure supplement 1*). Localization of this fusion protein was unaltered in *odr-1* or *tub-1* mutants (*Figure 5—figure supplement 1*). We, therefore, considered it unlikely that differential localization of INPP-1 accounts for $PI(4,5)P_2$ distribution in wild-type or *odr-1* mutant AWB cilia, although we cannot rule out the presence of additional $PI(4,5)P_2$ 5-phosphatases or spatially restricted modulation of INPP-1 activity (*Coon et al., 2012*; *Luo et al., 2012*).

*C. elegans* encodes a single type I PIP5K homolog that is highly expressed in the nervous system (*Weinkove et al., 2008*; *Xu et al., 2016*; *Xu et al., 2007*). In AWB, a GFP::PPK-1 fusion protein injected at a low concentration was membrane-associated and exhibited a localization pattern that was similar to that of GFP::PLCδ1-PH (*Figure 5D–E*, also see *Figure 6A*). This fusion protein was absent in all examined AWB cilia and also exhibited maximal fluorescence intensity in a region approximately 2–3 μm proximal from the cilia base (*Figure 5D–E*, also see *Figure 6A–B*). In *odr-1* mutants, we observed significantly increased relative levels of GFP::PPK-1 in AWB cilia (*Figure 5D–E*, also see *Figure 6A*, *Figure 6C*). We conclude that in *odr-1* mutants, increased ciliary localization of PPK-1 contributes in part to increased ciliary levels of $PI(4,5)P_2$.

## TUB-1 regulates PPK-1 and $PI(4,5)P_2$ distribution in AWB dendrites

We next investigated the mechanisms by which PPK-1 distribution in the AWB dendrites and cilia is regulated in wild-type and *odr-1* mutant animals. Unexpectedly, we identified a role for TUB-1 in regulating PPK-1 localization. As shown in *Figure 6A–B*, as compared to PPK-1 distribution in wild-type animals, the majority of *tub-1* mutant animals exhibited a proximal shift in GFP::PPK-1 localization with maximal enrichment starting at >4 μm from the cilia base in AWB dendrites (*Figure 6A–B*). Consistent with the observed change in GFP::PPK-1 distribution, GFP::PLCδ1-PH distribution in the AWB dendritic membrane was also altered upon loss of *tub-1*. In *tub-1* mutants, localization of this fusion protein shifted proximally in the AWB dendrites with maximal fluorescence intensity at >4 μm from the cilia base (*Figure 6D–E*). This phenotype was partially rescued upon expression of full-length TUB-1 protein in AWB (*Figure 6—figure supplement 1*). We conclude that TUB-1 itself regulates dendritic phosphoinositide composition by regulating localization of PPK-1 in AWB.

If TUB-1-mediated ciliary localization of PPK-1 is necessary for sensory signaling-dependent redistribution of $PI(4,5)P_2$, we would expect that loss of *tub-1* would suppress the altered ciliary phosphoinositide phenotype of *odr-1* mutants. Consistent with this notion, we found that loss of *tub-1* significantly suppressed the increased ciliary $PI(4,5)P_2$ localization phenotype of *odr-1* mutants (*Figure 6D*, *Figure 6F*). Similarly, GFP::PPK-1 was no longer found in the cilia of *tub-1; odr-1* double mutants (*Figure 6A*, *Figure 6C*). However, the dendritic localization patterns of the $PI(4,5)P_2$ sensor and GFP::PPK-1 in AWB in *tub-1; odr-1* double mutants were not identical to those of *tub-1* single mutants alone. Few animals exhibited the proximal dendritic shift of GFP::PLCδ1-PH or GFP::PPK-1 localization observed in *tub-1* single mutants. Instead, both proteins were shifted towards the ciliary base, although excluded from the AWB cilia proper, in *tub-1; odr-1* double mutants (*Figure 6*). We interpret these results to suggest that TUB-1 is necessary to localize PPK-1 within AWB cilia in *odr-1* mutants, but that an alternative mechanism partly compensates for loss of TUB-1 to localize PPK-1 to the ciliary base in *tub-1; odr-1* double mutants.

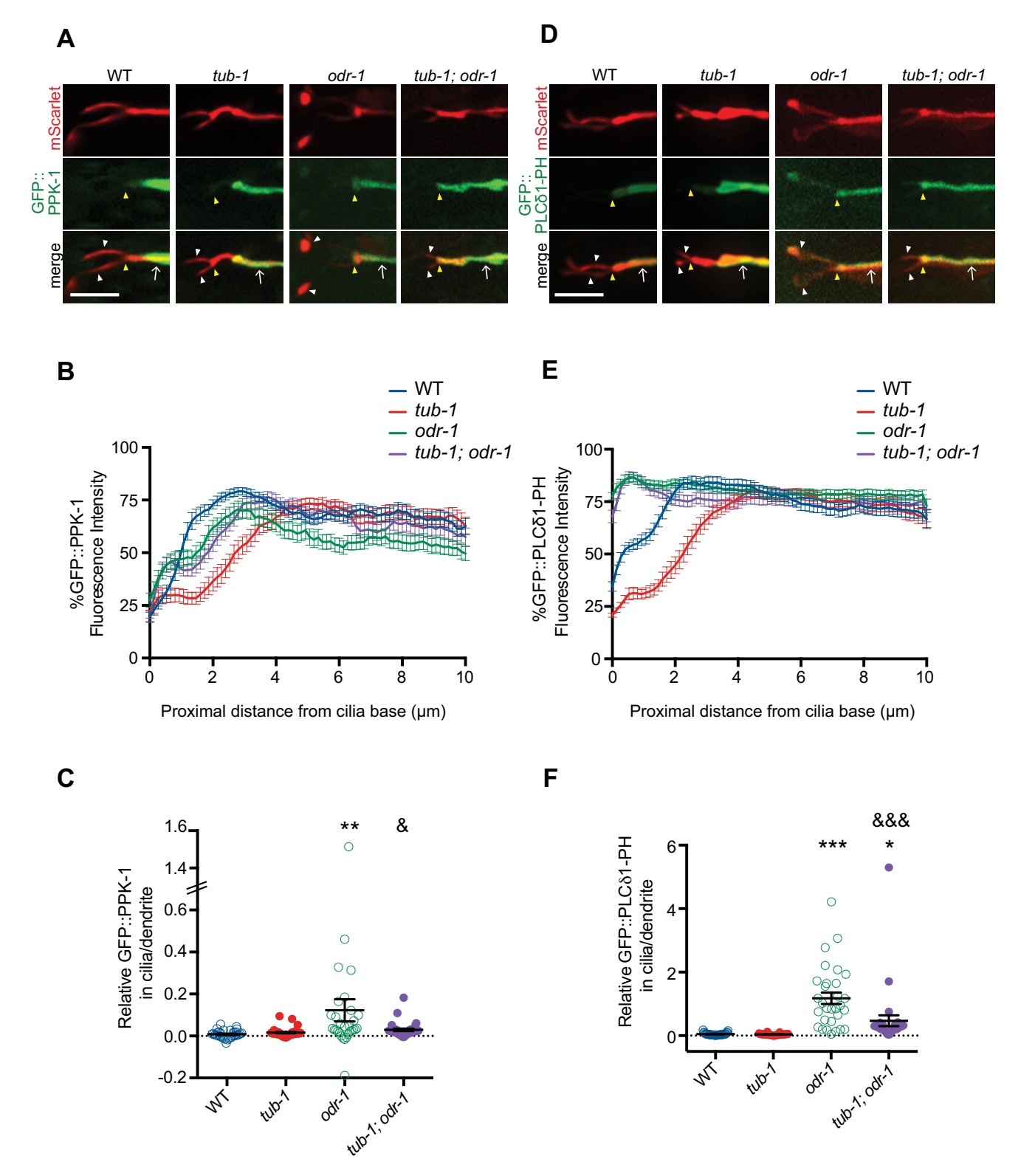

**Figure 6.** TUB-1 regulates PPK-1 localization and PI(4,5)P$_2$ distribution in AWB dendrites and cilia. (A, D) Representative images of GFP::PPK-1 (A) and GFP::PLCδ1-PH (D) localization in animals of the indicated genotypes. Alleles used were *tub-1(nr2004)* and *odr-1(n1936)*. The AWB neuron is marked via expression of *str-1*p::GFP::PPK-1::SL2::mScarlet or GFP::PLCδ1-PH::SL2::mScarlet (A, D, respectively). Yellow and white arrowheads indicate the cilia base and cilia, respectively; arrow indicates the dendrite. Scale bars: 5 μm. (B, E) Line scans of GFP::PPK-1 (B) and GFP::PLCδ1-PH (E) intensities in AWB

*Figure 6 continued*

dendrites of animals of the indicated genotypes. Zero indicates the cilia base/dendritic tip. Fluorescence intensities were normalized to the maximum intensity for each individual animal across the measured region, and the percent of this maximum intensity was calculated at each location to create individual line scans. Error bars are the SEM; n ≥ 30 neurons each. (C, F) Relative fluorescence intensities of GFP::PPK-1 (C) and GFP::PLCδ1-PH (F) in AWB cilia vs dendrites in animals of the indicated genotypes. Open circles indicate data repeated from *Figure 5*. *, ** and *** indicate different from wild-type at p<0.05, p<0.01, and p<0.001; & and &&& indicate different from *odr-1* at p<0.05 and p<0.001, (ANOVA and post-hoc Tukey's test). Each dot represents measurements from a single AWB neuron. Horizontal line is mean; error bars are SEM.

DOI: https://doi.org/10.7554/eLife.48789.016

The following source data and figure supplements are available for figure 6:

**Source data 1.** Data for *Figure 6B,C,E,F*, *Figure 6—figure supplement 1*, and *Figure 6—figure supplement 2B and C*.
DOI: https://doi.org/10.7554/eLife.48789.019

**Figure supplement 1.** TUB-1 may act cell-autonomously to regulate GFP::PLCδ1-PH distribution in AWB.
DOI: https://doi.org/10.7554/eLife.48789.017

**Figure supplement 2.** TUB-1 does not regulate PPK-1 and PI(4,5)P₂ distribution in ASK dendrites and cilia.
DOI: https://doi.org/10.7554/eLife.48789.018

Given the distinct roles of TUB-1 in wing and channel cilia, we asked whether TUB-1 also regulates ciliary $PI(4,5)P_2$ distribution in ASK. As in AWB, TUB-1 was enriched at the PCMC of wild-type ASK (*Figure 6—figure supplement 2A*). Moreover, similar to observations in AWB, both GFP::PLCδ1-PH and GFP::PPK-1 were excluded from ASK cilia and exhibited maximal fluorescence at a distance of >3 μm from the cilia base in wild-type animals (*Figure 6—figure supplement 2B–C*). We observed no changes in the distribution pattern of either protein in *tub-1* mutants (*Figure 6—figure supplement 2B–C*). We conclude that TUB-1 regulates localization of PPK-1 and $PI(4,5)P_2$ specifically in AWB but not in ASK.

## TUB-1 regulates PI(4,5)P₂ distribution in AWB dendrites in part via localization of the DPY-23 AP-2 μ2 subunit

We next asked how TUB-1 might regulate PPK-1 and $PI(4,5)P_2$ distribution in AWB dendrites. PIP5Ks are recruited to specific membrane subdomains via multiple mechanisms including direct interaction with AP-2 μ2 subunits (*Krauss et al., 2006*; *Ling et al., 2007*; *Nakano-Kobayashi et al., 2007*; *Padrón et al., 2003*). Tubby family proteins including *C. elegans* TUB-1 have previously been implicated in the regulation of endocytosis in multiple cellular contexts (*Chen et al., 2012*; *Mukhopadhyay et al., 2007a*; *Wahl et al., 2016*). Moreover, TUB interacts directly with Dynamin (*Xi et al., 2007*). We asked whether TUB-1 might regulate PPK-1 recruitment at the AWB PCMC in part via regulation of endocytic protein localization.

In wild-type animals, a DPY-23::GFP AP-2 μ2 subunit fusion protein was enriched at the PCMC of AWB in a domain similar to that occupied by TUB-1 (*Kaplan et al., 2012*) (*Figure 7A*). Levels of this fusion protein at the PCMC were significantly decreased although not abolished in *tub-1* mutants in AWB (*Figure 7A*). If DPY-23 plays a role in recruiting PPK-1, we would expect PPK-1 and $PI(4,5)P_2$ distribution in AWB to be affected in *dpy-23* mutants. Although we were unable to examine the distribution of GFP::PPK-1 due to technical reasons, the distribution pattern of GFP::PLCδ1-PH was shifted proximally in the AWB dendrite in *dpy-23* mutants, in a pattern similar but not identical to the distribution observed in *tub-1* mutants (*Figure 7B*). We observed no changes in the localization of a DYN-1::GFP dynamin fusion protein at the AWB PCMC in *tub-1* mutants (*Figure 7—figure supplement 1A*). A role for TUB-1 in regulating DPY-23 localization was specific for AWB, since localization of both DPY-23 and DYN-1 at the ASK PCMC was unaffected in *tub-1* mutants (*Figure 7—figure supplement 1B–C*). We conclude that disrupted localization of DPY-23 in *tub-1* mutants may in part contribute to the altered PPK-1 and $PI(4,5)P_2$ distribution at the AWB PCMC.

Although *tub-1* and *dpy-23* mutants exhibit related defects in $PI(4,5)P_2$ distribution in the AWB dendrite, their AWB cilia phenotypes are distinct. Unlike the truncated cilia phenotype of *tub-1* mutants, endocytic mutants exhibit expanded ciliary membraneous fans in AWB similar to the phenotypes of sensory signaling mutants (*Kaplan et al., 2012*) (*Figure 7C*). As in *odr-1* mutants, loss of *tub-1* fully suppressed the expanded ciliary membrane phenotype of *dpy-23* mutants (*Figure 7C*); AWB cilia lengths in *tub-1; dpy-23* mutants were similar to those in *tub-1* mutants alone (*Figure 7C*). However, in contrast to observations in *odr-1* animals in which ciliary TUB-1 levels are increased, we

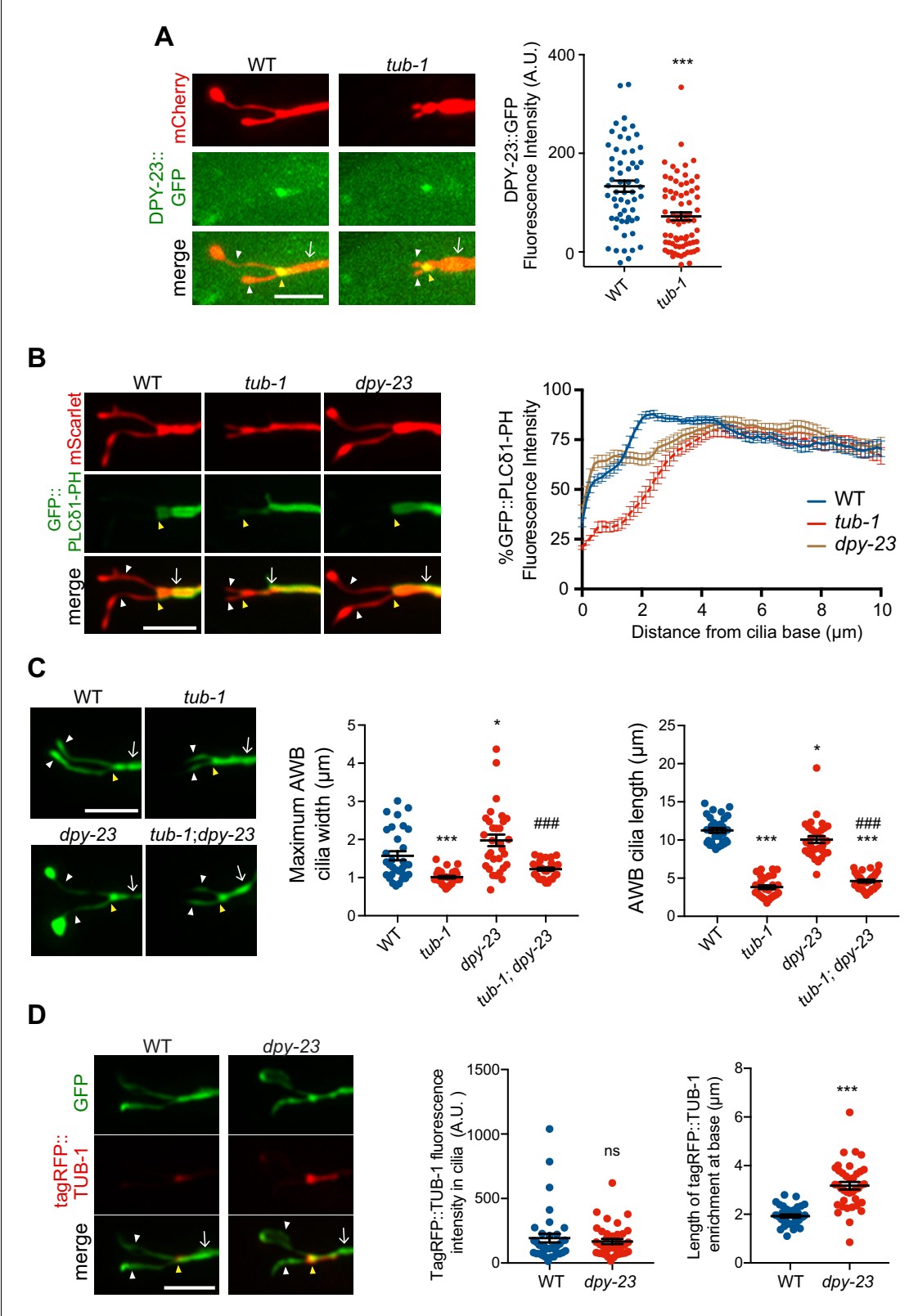

**Figure 7.** TUB-1 acts in part via the DPY-23 AP-2 μ subunit to localize PPK-1 at the PCMC. (A, B) (Left) Representative images of DPY-23::GFP (A) and GFP::PLCδ1-PH (B) localization in AWB cilia of animals of the indicated genotypes. AWB cilia were visualized via expression of *str-1*p::mCherry (A) or of *str-1*p::GFP::PLCδ1-PH::SL2::mScarlet (B). (A, Right) Quantification of DPY-23::GFP intensity in a 1 μm² area at the cilia base in A. *** indicates different from wild-type at p<0.001 (t-test). (B, right) Line scans of GFP::PLCδ1-PH intensity in AWB dendrites of animals of the indicated genotypes. Zero

*Figure 7 continued on next page*

*Figure 7 continued*

indicates the cilia base/dendritic tip. Dashed lines indicate data repeated from *Figure 6*. Error bars are the SEM; n ≥ 29 neurons each. Alleles used were *tub-1(nr2004)* and *dpy-23(e840)*. (C) (Left) Representative images of AWB cilia morphology in animals of the indicated genotypes. AWB cilia were visualized via expression of *str-1*p::*gfp*. (Middle) Quantification of total AWB cilia lengths and (right) maximum cilia widths of individual animals of the indicated genotypes. \*\*\* and ### indicate different from wild-type and *dpy-23(e840)* mutants, respectively, at p<0.001 (ANOVA with Tukey's post-hoc test). (D) (Left) Representative images of tagRFP::TUB-1 localization in AWB cilia of wild-type and *dpy-23(e840)* animals. AWB cilia were visualized via expression of *str-1*p::*gfp*. (Middle) Quantification of tagRFP::TUB-1 intensity within cilia, and (right) localization at the AWB cilia base in wild-type and *dpy-23(e840)* animals. \*\*\* indicates different from wild-type at p<0.001, ns – not significant (t- test). In all images, yellow and white arrowheads indicate the cilia base and cilia, respectively; arrow indicates the dendrite. Scale bars: 5 μm. In all scatter plots, each dot represents measurements from a single AWB neuron. Horizontal line is mean; error bars are SEM.

DOI: https://doi.org/10.7554/eLife.48789.020

The following source data and figure supplement are available for figure 7:

**Source data 1.** Data for *Figure 7A–D* and *Figure 7—figure supplement 1A–C*.
DOI: https://doi.org/10.7554/eLife.48789.022
**Figure supplement 1.** Localization patterns of endocytic proteins in AWB and ASK in *tub-1* mutants.
DOI: https://doi.org/10.7554/eLife.48789.021

observed no changes in ciliary concentrations of tagRFP::TUB-1 in *dpy-23* mutants (*Figure 7D*). Instead, this fusion protein occupied a larger area at the ciliary base in *dpy-23* animals (*Figure 7D*), similar to the mislocalization phenotype of a subset of ciliary IFT and transmembrane proteins in endocytic mutants (*Kaplan et al., 2012*). These observations imply that while TUB-1 is necessary for elaboration of the AWB ciliary membrane in wild-type, *odr-1,* and *dpy-23* mutants, the phosphoinositide and protein content of these fan-like structures are distinct in these mutant backgrounds.

## Discussion

We show here that the TUB-1 Tubby protein acts both at the PCMC and within cilia to regulate ciliary membrane morphogenesis, and ciliary and dendritic membrane phosphoinositide composition, in part via localization of the PPK-1 lipid kinase (*Figure 8*). Under normal growth conditions, TUB-1 excludes PPK-1, and thus $PI(4,5)P_2$, from cilia and regulates their distribution in AWB dendrites. This distribution may play an important role in regulating the correct balance between exocytosis and endocytosis at the PCMC, and trafficking of membrane proteins into cilia. In the absence of TUB-1, ciliary membrane proteins are mislocalized to the PCMC and the membraneous expansions of wing cilia are lost. TUB-1 is also necessary for sensory signaling-dependent remodeling of AWB cilia in *odr-1* mutants. In *odr-1* mutant animals, TUB-1 traffics PPK-1 into AWB cilia, thereby altering ciliary membrane phosphoinositide composition (*Figure 8*). Expanded cilia membrane volume together with increased ciliary signaling may represent a homeostatic mechanism that attempts to compensate for decreased sensory inputs in signaling mutants. We suggest that regulation of TUB-1-mediated transport represents a key mechanism underlying cell- and context-dependent ciliary morphological and functional diversification in a subset of *C. elegans* olfactory neurons.

How might TUB-1 regulate PPK-1 localization? The μ2 subunit of the AP-2 complex directly binds to and activates PIP5Ks (*Krauss et al., 2006*; *Nakano-Kobayashi et al., 2007*; *Posor et al., 2015*; *Wallroth and Haucke, 2018*). Localization of the DPY-23 AP-2 μ2 subunit at the AWB PCMC is decreased, although not abolished in *tub-1* mutants, suggesting that TUB-1 recruits PPK-1 at the PCMC in part via DPY-23 (*Figure 8*). Consistent with this hypothesis, the pattern of $PI(4,5)P_2$ distribution in AWB dendrites is similar in *tub-1* and *dpy-23* mutants. However, it is unlikely that TUB-1 acts via DPY-23 to increase PPK-1 levels within cilia in *odr-1* mutants. The small G protein ARF6 also binds and activates PIP5Ks in multiple contexts (*Funakoshi et al., 2011*; *Jones et al., 2000*). Several small GTPases including ARL13b have been implicated in ciliogenesis and ciliary membrane morphogenesis (*Blacque et al., 2018*; *Dilan et al., 2019*; *Li and Hu, 2011*; *Lu et al., 2015*), and TULP3/TUB regulates ciliary localization of ARL13b (*Han et al., 2019*; *Hwang et al., 2019*; *Legué and Liem, 2019*) (this work). Given a specific role of TUB-1 in localizing ARL-13 to AWB but not channel cilia, we speculate that TUB-1 may act via a related small GTPase in a cell-specific manner to increase PPK-1 and $PI(4,5)P_2$ levels in specialized cilia of sensory signaling mutants in *C. elegans*. However, it

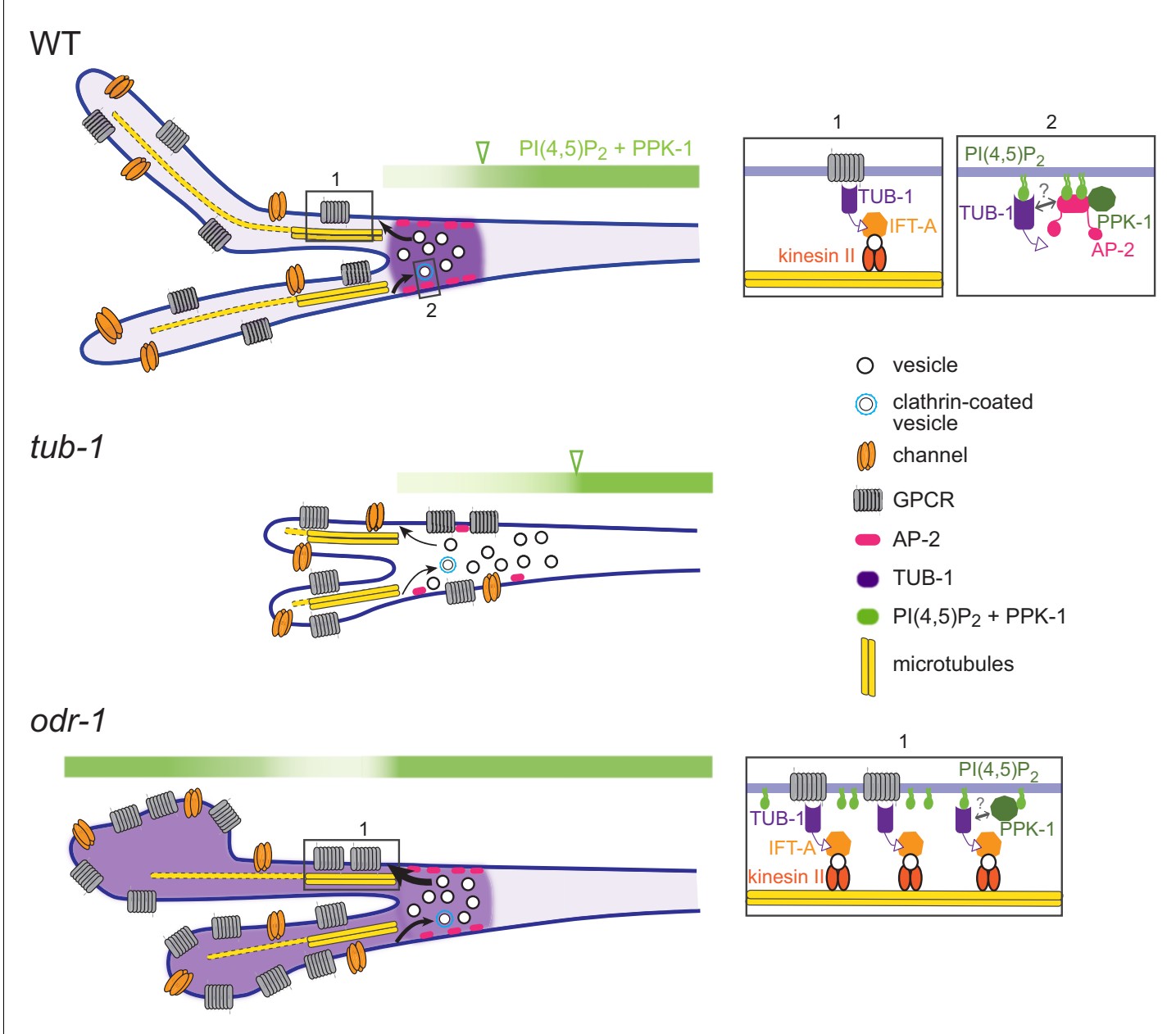

**Figure 8.** Model for TUB-1 function in regulating AWB olfactory cilia properties. Cartoons of proposed TUB-1 functions at the PCMC and within AWB cilia. Dotted yellow bars indicate singlet microtubules present in the distal segments of AWB cilia; the origins of these singlet microtubules are unclear (*Doroquez et al., 2014*). Numbered boxes in cartoons at left correspond to expanded diagrams at right. Open green arrowheads indicate the distal boundary of enrichment of PI(4,5)P_2 and PPK-1 in AWB dendrites. See text for additional details.

DOI: https://doi.org/10.7554/eLife.48789.023

is also possible that disruption of transition zone architecture in *odr-1* mutants enables ciliary access by PPK-1 (*Garcia-Gonzalo et al., 2011*; *Jensen et al., 2015*).

Altered PI(4,5)P_2 distribution at the PCMC in *tub-1* mutants is expected to significantly affect both endo- and exocytosis as well as entry and exit of ciliary proteins (*Grossman et al., 2011*; *Martin, 2012*; *Park et al., 2015*; *Posor et al., 2015*; *Wallroth and Haucke, 2018*). PI(4,5)P_2 levels also alter multiple aspects of ciliogenesis including transition zone maturation (*Gupta et al., 2018*; *Xu et al., 2016*; *Xu et al., 2019*). However, since initial steps of ciliogenesis appear to be unaffected in *tub-1* L1 larvae (see *Figure 1B*), we favor the hypothesis that TUB-1 regulates membrane

biogenesis in AWB via regulation of vesicular trafficking at the cilia base and membrane transport within cilia. TUB-1 was previously suggested to target the RAB-7 small GTPase implicated in endocytic trafficking in *C. elegans* (*Mukhopadhyay et al., 2007a*), and Tubby proteins have similarly been implicated in the regulation of both exo- and endocytosis in multiple contexts including in cilia and photoreceptor ribbon synapses (*Chen et al., 2012*; *Grossman et al., 2011*; *Hagstrom et al., 2001*; *Hagstrom et al., 1999*; *Park et al., 2013*; *Wahl et al., 2016*). Our results raise the possibility that TULP1/dTULP modulates these processes via regulation of local PI(4,5)P$_2$ pools.

The distribution and function of transmembrane signaling proteins is highly regulated by membrane lipids (*Balla, 2013*; *Gamper and Shapiro, 2007*; *Hansen, 2015*; *Yen et al., 2018*). Consistently, ciliary membrane lipid composition regulates targeting of ciliary transmembrane proteins including those required for Shh signaling, mechanotransduction, and phototransduction (*Bae et al., 2009*; *Boesze-Battaglia and Schimmel, 1997*; *Chávez et al., 2015*; *Garcia-Gonzalo et al., 2015*; *Giusto et al., 2010*; *Park et al., 2015*; *Raleigh et al., 2018*; *Tyler et al., 2009*). In *C. elegans*, under conditions of decreased sensory input as in *odr-1* mutants, increasing ciliary surface to volume ratio together with alterations in ciliary PI(4,5)P$_2$ levels may increase sensitivity to external stimuli. Do TULP/TUB proteins also regulate PI(4,5)P$_2$ distribution within mammalian cilia? Although to our knowledge ciliary PI(4,5)P$_2$ levels in TULP/TUB mutants have not been directly examined in mammalian cells, recent observations indicate that TULP3/TUB regulate ciliary enrichment of INPP5e via localization of ARL13b (*Han et al., 2019*; *Humbert et al., 2012*; *Hwang et al., 2019*; *Legué and Liem, 2019*). Based on the phenotype of *inpp5e* mutants, altered localization of INPP5e in TULP3/TUB mutants would be expected to increase ciliary membrane PI(4,5)P$_2$ levels (*Chávez et al., 2015*; *Garcia-Gonzalo et al., 2015*; *Park et al., 2015*), suggesting a conserved role for Tubby proteins in regulating membrane phosphoinositide composition. These observations raise the intriguing possibility that dynamic TULP3/TUB-mediated ciliary transport of lipid-modification enzymes may represent a conserved mechanism that mediates plasticity in ciliary membrane protein composition and function in multiple contexts.

Although TUB-1 is necessary for membrane expansion in both *odr-1* and *dpy-23* mutants, increased ciliary localization of TUB-1 and/or PI(4,5)P$_2$ is neither necessary nor sufficient for ciliary membrane expansion. The expanded AWB ciliary membrane in *dpy-23* mutants does not contain increased TUB-1 or PI(4,5)P$_2$, and increasing TUB-1 and PI(4,5)P$_2$ within cilia does not lead to membrane expansion in *inpp-1* mutants. In *odr-1* mutants, increased TUB-1-mediated transport of membrane cargoes and PPK-1 to cilia may cause expansion of the ciliary membrane together with altered membrane phosphoinositide composition (*Figure 8*). In this model, while ciliary PI(4,5)P$_2$ is also increased in *inpp-1* mutants, TUB-1-mediated cargo delivery to cilia may not be enhanced similarly to *odr-1* mutants, resulting in a failure to enlarge the cilia membrane. In endocytic mutants, while TUB-1 function at the PCMC and cilia continues to be necessary for membrane biogenesis, disruption of membrane retrieval mechanisms relative to membrane delivery results in AWB membrane volume remodeling (*Kaplan et al., 2012*). Regulation of ciliary membrane volume, lipid composition and protein content via multiple independent mechanisms may provide additional flexibility in reshaping cilia structure and function (*Garcia et al., 2018*; *Lechtreck et al., 2013*; *Nachury, 2018*; *Phua et al., 2017*). It will be particularly interesting to determine whether distinct mechanisms operate to modulate cilia morphology in different sensory signaling regimes.

Dynamic modulation of intracellular trafficking and cellular structures is a hallmark of cellular functional plasticity (*Jin and Qi, 2018*; *Jontes and Smith, 2000*; *Nakahata and Yasuda, 2018*; *Sugie et al., 2018*). This plasticity is particularly critical in neurons which must continuously modulate their properties in response to the animal's experience and environment. Sensory cilia are subject to similar experience-dependent plasticity in their structures and signaling functions (*Besschetnova et al., 2010*; *Caspary et al., 2007*; *Mesland et al., 1980*; *Mukhopadhyay et al., 2008*); this work implicates the conserved TUB-1 protein in mediating this plasticity. An important open issue for the future will be to establish how sensory signals are interpreted and translated to alter TUB-1 properties in wing cilia, and the consequences of ciliary remodeling on neuronal functions.

## Materials and methods

### C. elegans genetics

Wild-type animals were *C. elegans* variety Bristol, strain N2. *C. elegans* strains were grown at 20°C on standard nematode growth media (NGM) plates seeded with the *Escherichia coli* strain OP50. Standard genetic techniques were used to introduce transgenes into mutant genetic backgrounds. Transgenic animals were generated by microinjection of test plasmid(s) at 0.5 ng–50 ng/µl of plasmid(s) together with a co-injection marker (*unc-122*p::*gfp*, *unc-122*p::*mCherry*, or *unc-122*p::*dsRed*) at 50 ng/µl. Mutations were confirmed by PCR and/or sequencing. Except where indicated, the same transgenic array was examined in both wild-type and mutant backgrounds. Animals from at least two independent lines were examined for each transgenic strain, and one representative line was typically chosen for further analysis. A complete list of all strains used in this work is provided in *Supplementary file 1*.

### Molecular biology

Constructs driving expression specifically in AWB were generated by fusing relevant cDNAs to 3.0 kb *str-1* (*Troemel et al., 1997*) or the ~3.0 kb *srd-23* (*Colosimo et al., 2004*) upstream regulatory sequences. Constructs driving expression specifically in ASK were generated by fusing relevant cDNAs to 1.9 kb *srbc-64*, 1.6 kb *srbc-66* (*Kim et al., 2009*), or 2.9 kb *sra-9* (*Troemel et al., 1995*) upstream regulatory sequences. The *arl-13* cDNA was expressed in ASH under 3.8 kb *sra-6* (*Troemel et al., 1995*) upstream regulatory sequences. *tub-1*, *inpp-1* (isoform a), *dyn-1* (isoform b), *dpy-23* (isoform b), and *ppk-1* cDNAs were amplified from a mixed stage N2 cDNA library using gene-specific primers. *tax-2* and *tax-4*-expression constructs have been reported previously (*Wojtyniak et al., 2013*). The PLCδ1-PH-encoding cDNA was a gift from A. Rodal (Brandeis University). Human *Tulp1* and *Tulp3* cDNAs were a gift from S. Mukhopadhyay (UT Southwestern Medical Center). TUB-1 N- and C-terminal domain-encoding sequences were identified based on homology with *Mus musculus* Tubby sequences. All sequences were cloned into pPD95.77 (gift of A. Fire) or pMC10 (gift of M. Colosimo). N- and C-terminal reporter-tagged constructs were generated by subcloning fluorescent reporter sequences in-frame into expression vectors containing the gene of interest. In some cases, SL2::mScarlet coding sequences were also inserted to visualize neuron morphology. Point mutations in the *tub-1* cDNA were generated using site-directed mutagenesis. All constructs were confirmed by sequencing. A full list of plasmids used in this work is provided in *Supplementary file 2*.

### Imaging and image analyses

Except where indicated, all hermaphrodites were imaged as one day old adults, and one neuron per animal was examined. L1 larvae imaged in *Figure 1B* were obtained from growth synchronized adults, and imaged at 3–5 hr after hatch. All animals were mounted on 10% agarose pads set on microscope slides and immobilized using 10 mM tetramisole (Sigma). Unless noted otherwise, all imaging was performed on an inverted spinning disk confocal microscope (Zeiss Axiovert with a Yokogawa CSU22 spinning disk confocal head and a Photometerics Quantum SC 512 camera). Optical sections were acquired at 0.27 µm intervals using a 100X oil immersion objective and SlideBook 6.0 software (Intelligent Imaging Innovations, 3i). Images in *Figure 4A* were acquired at 0.25 µm intervals using a 63X oil immersion objective on an upright microscope (Zeiss Imager.M2 with a Hamamatsu C4742.95 camera) and were collected using Zeiss Zen software. Images in *Figures 5D* and *6A–C* were acquired at 0.3 µm intervals using a 100X oil immersion objective on an upright spinning disk microscope (Nikon Ni-E with a Yokogawa CSU-W1 spinning disk head and an Andor iXon 897U EMCCD camera) and were collected using Nikon Elements AR software. Optical sections were *z*-projected at maximum intensity using SlideBook 6.0 software or FIJI/ImageJ [National Institutes of health (NIH), Bethesda, MD].

Fluorescence microscopy image processing and analyses were performed using FIJI. All quantifications were performed using data from a minimum of two and typically three biologically independent experiments performed on independent days. Methods used for specific analyses are described below.

### Cilia length
Line segments were drawn from the cilia tip to the base and lengths of both AWB cilia per neuron were summed.

### Maximum cilia width
Cilium widths were quantified using line segments to identify the widest point of each AWB cilium, and the summed total was calculated.

### Cilia protein fluorescence intensities
Protein fluorescence intensities were quantified by outlining the cilia area in 2D maximum projected images and measuring the mean intensity. The average of three mean background intensities was subtracted from the mean intensity to arrive at the final value.

### Dendrite and base protein fluorescence intensities
Fluorescence intensities in dendrites were quantified by outlining a 0.5 µm area region of the dendrite 5–8 µm from the cilia base and measuring the mean fluorescence intensity after background subtraction. DPY-23, DYN-1, or TUB-1 fusion protein fluorescence intensities at the cilia base were quantified by drawing a 1 µm diameter circular region of interest at the cilia base in the 2D maximum projected images and measuring the mean fluorescence intensity after background subtraction.

### Ratios of cilia/dendrite fluorescence intensities
Protein fluorescence intensity ratios were analyzed in some instances to control for known expression differences driven by specific promoters such as *str-1* and *srd-23* in *odr-1* mutants (*van der Linden et al., 2007*; *van der Linden et al., 2008*). Ratios were calculated by normalizing cilia protein fluorescence intensities by dendritic or base protein fluorescence intensities (calculated as described above).

### Length of protein enrichment at the cilia base
To measure the length of TUB-1 enrichment at the cilia base, line segments were drawn from cilia base to the proximal end of the enrichment domain in the dendrite.

### Line scans
Line scans were generated by drawing lines from the ciliary base and along the dendrite, and measuring fluorescence intensities along the line using the plot profile tool in FIJI. Fluorescence intensities were normalized to the maximum intensity for each individual animal across the measured region, and the percent of this maximum intensity was quantified at each location to create individual line scans.

## Fluorescence recovery after photobleaching (FRAP)
Animals were mounted on 10% agarose pads and immobilized using 10 mM tetramisole. Animals were imaged at 0.6 µm interval z-stacks using a 100X oil immersion objective on an upright spinning disk microscope (Nikon Ni-E with a Yokogawa CSU-W1 spinning disk head and an Andor iXon 897U EMCCD camera). Images were collected using Nikon Elements AR software. Cilia were photobleached using a 405 nm laser (at 40% power), directed by an Andor Mosaic three digital micromirror device. One or both AWB cilia were photobleached in wild-type and *odr-1* mutants. Cilia were imaged at least 12 s prior to bleaching, and up to 2 min following the bleaching event at 3 s intervals to assess fluorescence recovery. Images were corrected for photobleaching using the Bleach Correction plugin and Simple Ratio Method in FIJI/Image J [National Institutes of health (NIH), Bethesda, MD]. Pre-bleach fluorescence was normalized to 100% in order to calculate the fraction of fluorescence recovery. The recovery half-times ($t_{1/2}$) and mobility fractions ($M_f$) were calculated using Prism 6 Software (Graphpad, La Jolla, CA) by fitting individual recovery curves using one phase association nonlinear regression. The mean fluorescence recovery curves were created by plotting the mean and

SEM of fluorescence intensities at individual time points after bleaching using Prism 6 Software (Graphpad, La Jolla, CA).

## Statistics

All plots were generated using Prism seven software. All scatterplots show mean ± SEM. Statistical significances among multiple strains were calculated using one-way ANOVA followed by Tukey's multiple comparisons test. For comparisons between two groups, an unpaired student's $t$ test with equal SD was used.

## Acknowledgements

We are grateful to the *Caenorhabditis* Genetics Center and Shohei Mitani (NRBP, Japan) for strains, Avital Rodal and Saikat Mukhopadhyay for reagents, and Avital Rodal, Steven Del Signore and Ashish Maurya for advice regarding FRAP. We thank the Sengupta lab and Saikat Mukhopadhyay for advice and comments on the work and manuscript. We are particularly grateful to Inna Nechipurenko for the illustration shown in *Figure 8*. This work was supported in part by the NIH (R35 GM122463 – PS.; T32 NS007292 – AP; T32 GM007122 – DD).

## Additional information

### Competing interests

Piali Sengupta: Reviewing editor, *eLife*. The other authors declare that no competing interests exist.

### Funding

| Funder | Grant reference number | Author |
| --- | --- | --- |
| National Institute of General Medical Sciences | R35 GM122463 | Piali Sengupta |
| National Institute of Neurological Disorders and Stroke | T32 NS007292 | Alison Philbrook |
| National Institute of General Medical Sciences | T32 GM007122 | Danielle DiTirro |

The funders had no role in study design, data collection and interpretation, or the decision to submit the work for publication.

### Author contributions

Danielle DiTirro, Alison Philbrook, Conceptualization, Data curation, Formal analysis, Validation, Investigation, Visualization, Methodology, Writing—review and editing; Kendrick Rubino, Investigation, Methodology; Piali Sengupta, Conceptualization, Supervision, Funding acquisition, Writing—original draft, Project administration, Writing—review and editing

### Author ORCIDs

Danielle DiTirro ⬤ https://orcid.org/0000-0003-0180-7634
Alison Philbrook ⬤ https://orcid.org/0000-0003-3330-3086
Piali Sengupta ⬤ https://orcid.org/0000-0001-7468-0035

### Decision letter and Author response

Decision letter https://doi.org/10.7554/eLife.48789.028
Author response https://doi.org/10.7554/eLife.48789.029

## Additional files

### Supplementary files

• Supplementary file 1. List of strains used in this work.

DOI: https://doi.org/10.7554/eLife.48789.024

• Supplementary file 2. List of plasmids used in this work.
DOI: https://doi.org/10.7554/eLife.48789.025
• Transparent reporting form
DOI: https://doi.org/10.7554/eLife.48789.026

## Data availability

All data generated or analyzed during this study are included in the manuscript and supporting files.

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
