## [Decision Letter]

Congratulations, we are pleased to inform you that your article, "The *C. elegans* Tubby homolog dynamically modulates olfactory cilia membrane morphogenesis and phospholipid composition", has been accepted for publication in *eLife*.

As you will see, the reviewers were very impressed by the work described in the manuscript, including rigor and clarity. One reviewer had a comment that you can consider when uploading the final version of the manuscript.

*Reviewer #1:*

The manuscript by DiTirro, Philbrook et al. examines the function of the Tubby homolog TUB-1 in ciliary membrane morphology of the *C. elegans* amphid wing AWB cilia. Authors convincing show that TUB-1 regulates morphology in a cell-type dependent manner (AWB but not ASK) and ciliary protein localization (TAX-4 and ARL-13 but not transition zone or IFT components). Next, they identify ciliary targeting determinants (IFT-A binding sites in the N-terminal domain). To link structure with function, authors show that *tub-1* is required for sensory signaling ciliary membrane expansion (in *odr-1* receptor guanylyl cyclase mutant). PIP2 and the PIP5K PPK-1 are normally excluded from AWB cilia but is present in the *odr-1* mutant. From here, authors show that TUB-1 regulates PIP2 distribution via PPK-1 localization but not the localization of the INPP5e phosphatase INPP-1. Finally, authors connects PIP2 and the AP-2 μ2 endocytic adaptor DPY-23. The model for TUB-1 function in AWB is comprehensive and supported by strong data. This is a logically flowing and easy to read manuscript that will appeal to the readership of *eLife*.

While not the point of the experiment, it is interesting that ARL-13 localizes along the entire length of the AWB cilium. In amphid channel cilia, ARL-13 is restricted to the doublet/middle segment. Could authors comment on this?

*Reviewer #2:*

In this paper, the authors report on the functions of a *C. elegans* Tubby homolog, TUB-1, in controlling the shape of sensory neuron cilia, and localization of sensory proteins within these compartments. The authors reveal that TUB-1 is required for both initial, and activity-dependent formation of ciliary elaborations, and that this is mediated in part by trafficking of a PIP5 kinase. Consistent with this, changes in phosphoinositides in ciliary membranes are documented.

Cilia are ubiquitous structures in animals, and are thought to adorn most mammalian cells. In general, these structures serve as signaling compartments, and understanding how these structures are formed and signaling proteins localized within them has broad implications for cell-cell signaling in the nervous system and outside the nervous system.

The studies presented here are outstanding in rigor and clarity. The authors' interpretations of their results are plausible and consistent with other observations in the field, but provide a step forward in implicating TUB-1 in neuronal plasticity. In my opinion, the paper is fine as is.

Nonetheless, if the authors are inclined, it would be interesting to understand more about the effects of *tub-1* mutations on animal behavior. Are responses to environmental odors altered? Are these alterations consistent with the cell biological defects?

---

## [Author Response]

Reviewer #1:[…]While not the point of the experiment, it is interesting that ARL-13 localizes along the entire length of the AWB cilium. In amphid channel cilia, ARL-13 is restricted to the doublet/middle segment. Could authors comment on this?

As the reviewer notes, the localization pattern of ARL-13 is indeed distinct between the AWB and channel cilia (first reported by Cevik et al., 2010). Since we do not currently know the reason or the mechanism underlying this cell-specific difference in subcellular localization pattern, we prefer to not speculate at this time.

Reviewer #2:[…]Nonetheless, if the authors are inclined, it would be interesting to understand more about the effects of tub-1 mutations on animal behavior. Are responses to environmental odors altered? Are these alterations consistent with the cell biological defects?

A previous report showed that *tub-1* mutants exhibit defects in their responses to a subset of volatile attractive odorants, including chemicals known to be sensed by AWC and AWA (Mak et al., 2006). We plan to follow up on these observations in the future with more detailed comparisons and analyses of the effects of *tub-1* mutations on both odorant-evoked behaviors as well as neuronal activity in sensory neurons with ‘wing’ and ‘channel’ cilia.